# First-in-Human Study to Investigate the Safety Assessment of Peri-Implant Soft Tissue Regeneration with Micronized-Gingival Connective Tissue: A Pilot Case Series Study

**DOI:** 10.3390/medicines10010009

**Published:** 2023-01-04

**Authors:** Takashi I, Sawako Noda, Seigo Ohba, Izumi Asahina, Yoshinori Sumita

**Affiliations:** 1Department of Regenerative Oral Surgery, Nagasaki University Graduate School of Biomedical Sciences, 1-7-1 Sakamoto, Nagasaki 852-8588, Japan; 2Department of Medical Research and Development for Oral Disease, Nagasaki University Graduate School of Biomedical Sciences, 1-7-1 Sakamoto, Nagasaki 852-8588, Japan

**Keywords:** gingiva, connective tissues, clinical study, regeneration

## Abstract

**Background:** We have recently proposed an alternative strategy of free gingival graft (FGG) and connective tissue graft (CTG) using micronized-gingival connective tissues (MGCTs). The advantage of this strategy is that MGCTs from a small piece of maxillary tuberosity can regenerate the keratinized tissue band. However, safety and efficacy have not yet been established in patients. This clinical study was a pilot case series, and the objective was to assess the safety and the preliminary efficacy of MGCTs on peri-implant mucosa regeneration. **Methods:** This was a pilot interventional, single-center, first-in-human (FIH), open (no masking), uncontrolled, and single-assignment study. A total of 4 patients who needed peri-implant soft tissues reconstruction around dental implants received transplantation of atelocollagen-matrix with MGCTs micronized by the tissue disruptor technique. The duration of intervention was 4 weeks after surgery. **Results:** This first clinical study demonstrated that using MGCTs did not cause any irreversible adverse events, and it showed the preliminary efficacy for peri-implant soft tissues reconstruction in dental implant therapy. **Conclusions:** Though further studies are needed on an appropriate scale, as an alternative strategy of FGG or CTG, MGCTs might be promising for peri-implant mucosa reconstruction without requiring a high level of skills and morbidity to harvest graft tissues.

## 1. Introduction

Successful treatments in dental implants require an adequate keratinized tissue band to maintain peri-implant health, since an insufficient keratinized tissue band results in peri-implant mucositis, unsatisfactory esthetic outcomes, and ultimately lower survival rates of dental implants [1,2,3]. Meanwhile, tooth extraction can lead to alveolar bone atrophy and concurrent decline in keratinized tissue band. Therefore, reconstruction or augmentation of adequate peri-implant mucosa is indispensable for successful treatment in dental implants [4,5,6]. In clinical practice, the free gingival graft (FGG) and the connective tissue graft (CTG) are currently the gold standards for augmenting keratinized tissue [2,7,8,9,10]. The FGG and the CTG have been used to re-establish the keratinized tissue, correcting mucogingival deformities, and improving the esthetics at teeth and dental implant sites. However, these procedures include some disadvantages: inability to harvest large grafts; high morbidity rate after surgery; and inadequate width and thickness of peri-implant mucosa at the recipient site [7,11,12,13]. Therefore, an alternative strategy that avoids donor-site morbidity but has comparable healing potential is needed.

To date, many keratinized oral mucosa have been engineered utilizing allo- or xeno-genic materials, such as acellular human dermal matrix (ADM), and porcine collagen matrix (CM). ADM and CM have shown utility in augmenting the keratinized mucosa without donor-site morbidity. However, they could not provide sufficient width and thickness of peri-implant mucosa owing to their high levels of matrix contraction [8]. Meanwhile, mesenchymal stromal cells (MSCs) have identified numerous mechanisms by which these cells can attenuate the acute inflammatory response, and promote re-epithelialization and extracellular matrix remodeling via their paracrine mechanisms [14,15]. More specifically, gingival mucosa-derived mesenchymal stem cells (GMMSCs) have shown to promote wound healing [16,17,18,19,20]. In fact, several studies have revealed that GMMSCs possess multipotent differentiation capacities and potent immunomodulatory effects, providing various types of beneficial factors with immunosuppressive and anti-inflammatory functions [21,22,23]. These properties have endowed GMMSCs with regenerative and therapeutic potentials in the treatment of peri-implant mucosa defects. On the other hand, Izumi et al. reported that ex vivo–produced oral mucosa equivalent (EVPOME) generated a keratinized mucosal surface epithelium in a clinical trial [24]. The EVPOME was made from AlloDerm (an acellular dermal matrix; LifeCell) seeded on primary oral keratinocytes, harvested from palatal keratinized mucosa. However, in the clinical use of GMMSCs or keratinocytes, obtaining sufficient numbers of cells may be challenging because an expansion process is required in vitro. In general, these processes of cultivation require time, labor, and costs related to the tissue harvest and the cell culture. Therefore, there remains a need for an alternative strategy that avoids donor-site morbidity, but provides sufficient reconstruction of peri-implant mucosa in dental implant therapy.

Recently, new possibilities for minimally invasive and non-laborious procedures using micronized tissues (micro-graft) approximately 50 μm in size from small autologous tissue samples (only a few millimeters in size) can be used effectively for bone or soft tissue engineering. The medical device, the tissue disruptor technique using the Rigenera^®^ machine and Rigeneracons^®^ (Human Brain Wave, Turin, Italy), provides autologous micro-grafts immediately available for use in clinical practice. The created micro-grafts from different kinds of human tissues, such as dental pulp, adipose, or dermal tissues, have shown high cell viability and optimal regenerative and differentiation potential [25,26,27,28,29,30,31,32]. In particular, micro-grafts from adipose or dermal connective tissues have shown to improve wound healing, promoting the re-epithelialization and the softness of the tissue. Flow cytometric analysis demonstrated that the micro-grafts contained abundant MSCs [25,28,29,33]. Therefore, we focused on an alternative strategy using micronized tissue transplantation (micro-grafts) and found that transplantation of porcine micronized-gingival connective tissues (MGCTs) using the tissue disruptor technique promoted skin wound healing in immune-deficient mice [34]. The MGCTs contained gingival mucosa-derived mesenchymal stem cells (GMMSCs), which are known to promote wound healing with superior scar quality [35]. In fact, mesenchymal cells derived from MGCTs were detected mainly in the matrix immediately below the epithelialization area. The expression of mRNAs related to endothelial and epithelial cell marker genes (*cd31, vWF, cytokeratin, vegf-a, vegf-b, flt-1,* and *flk-1*) were elevated in transplanted samples. Additionally, our data using proteome profile array showed the increased expression levels of proangiogenic factors, such as Angiogenin, EGF, FGF, IGF, IP-10, KC, MMP-3, -9, SDF-1, and VEGF, in MGCTs. Therefore, survived mesenchymal cells derived from MGCTs promoted vascularization and re-epithelialization in a paracrine manner. Moreover, while a small number of MGCT derived cells may have contributed directly to form the blood vessels or epidermis in the wound area, the majority of donor cells likely functioned as mesenchymal cells. Then, the advantage of this strategy is that the MGCTs from a small piece (approximately 3 mm^3^) of maxillary tuberosity can regenerate the skin wound approximately 20-fold as large as harvested tissue. Our findings suggested that this strategy could easily be applied to the clinical setting, showing low morbidity and ready availability, and it could contribute to peri-implant soft tissues reconstruction or augmentation using the tissue disruptor technique.

The micro-graft with MGCTs has a potential as an alternative therapy to the conventional FGG or CTG in the clinical setting because of its low morbidity and ready availability. However, there is no information regarding micro-grafts with MGCTs from a clinical study. Therefore, the safety and preliminary efficacy of micro-grafts with MGCTs for peri-implant soft tissues regeneration in dental implant treatment were evaluated in this pilot case series.

## 2. Materials and Methods

### 2.1. Study Design

The protocol of this clinical study was approved by The Clinical Research Review Board of Nagasaki University (approval no. CRB7180001) and registered with the Japan Registry of Clinical Trials (trial no. jRCTs072180075) and the University Hospital Medical Information Network (UHMIN) (trial no. UMIN000027150) as “Keratinized gingival tissue regeneration with micronized-gingival connective tissues: Pilot Study (Keratinized tissue reconstruction with gingival micro-graft)”. The study was performed in accordance with the Helsinki Declaration. All participants provided written, informed consent to participate in the study. The study was an interventional, single-center (Nagasaki University Hospital), single-arm, open (no masking), uncontrolled, single-assignment study. The primary endpoint was safety/tolerability. In part, the preliminary efficacy of peri-implant soft tissues reconstruction was assessed. This study was conducted between March 2017 and May 2020. The participants were enrolled according to the following inclusion criteria and interviewed regarding their underlying health conditions and blood tests before the operation. Four patients who signed the approved consent form were included in this study. The procedure of this pilot case series is summarized in Figure 1.


**Patient Inclusion Criteria**



**Dental implant treatment.**

**Keratinized tissue band defect or insufficient thickness (<2 mm).**

**Oral health care and good plaque control.**

**Aged 20 years or older, but younger than 75 years.**

**Written, informed consent (IC) obtained from the patient him/herself.**

**Intention and ability to visit a hospital.**

**Intention to use barrier contraception during the study period.**



**Patient Exclusion Criteria**



**Any malignancy or sepsis.**

**Severe autoimmune or endocrinological disease; coagulation abnormality (PT < 50% or outside the APTT range 23.5–42.5 s).**

**Syphilis test/HBV antigen/HCV antigen/anti-HTLV-1 antibody/anti-HIV antibody positivity.**

**Liver dysfunction (two biomarkers of liver function with concentrations outside the following range: AST 10–40 IU/L, ALT 5–45 IU/L).**

**Pregnancy.**

**Risk of allergy to the drug used in this study.**

**Transmissible spongiform encephalopathy.**

**Dementia.**

**Metabolic bone disease or treated with bisphosphonates.**

**Smoking habit.**

**Severe periodontitis.**

**Judged by the clinical investigator as inappropriate for this study.**


### 2.2. Preparation of the Micro-Graft with Micronized-Gingival Connective Tissue

The procedures for preparing the micro-graft are presented in Figure 2. A piece (approximately 3 mm^3^) of gingival connective tissue was harvested from the palatal mucosa of the maxillary tuberosity. Then, the harvested tissue was mechanically dissociated immediately with the tissue disruptor technique using the Rigenera^®^ machine and Rigeneracons^®^, a mechanical disruptor of small tissues (Human Brain Wave, Turin, Italy), according to a method previously described [26,27,29,34]. With 1 mL of saline solution per piece, the harvested tissue was micronized for 90 s by the tissue disruptor machine. In using the device, all processes are fully automatic. Then, micronized-gingival connective tissues (MGCTs, ≤100 μm each) were suspended in the saline and collected into a 1.5-mL sterilized tube. Subsequently, the suspended micronized-gingival connective tissue was seeded onto a square shaped atelocollagen-matrix (TERUDERMIS^®^; Olympus Terumo Biomaterials, Tokyo, Japan), which was put with the silicone membrane face-down on the sterilized dish just before the transplantation. An atelocollagen-matrix was adjusted for suitable size of the exposed periosteum by handling with straight tweezers and surgical scissors before seeding.

### 2.3. Surgical Procedure

The surgical procedure was performed with local anesthesia in an operating room, as shown in Figure 3. In both one- and two-stage dental implants, the peri-implant mucosa surrounding the implants was shifted apically with the remaining periosteum. The exposed periosteum was covered by the micro-graft with MGCT, and the collagen side was put face down onto the wound surface (with the silicone membrane). Subsequently, the circumference of the transplanted micro-graft was fixed to the edges of the wound using non-absorbable thread (Nylon) with locked single-layer suturing techniques. After applying it to the wound, a periodontal dressing was placed.

### 2.4. Outcome Measurements

Study visits took place at baseline (approximately 1 month before the surgery) and on specified days after the surgery. Wound irrigation was performed with saline 1 day and 1 week after the surgery, and the periodontal dressing and sutures were removed between 1 and 2 weeks after surgery. Four weeks after the surgery, the final safety assessment was performed, and this pilot study was completed. In addition, the reconstruction of peri-implant soft tissues was finally assessed 16 weeks after surgery. A physical examination including the oral cavity and a safety assessment were performed at every visit. The primary endpoint was the safety/tolerability of the protocol. The safety and tolerability were assessed by the presence or the absence of an adverse events (AEs). The safety assessment was conducted by an independent safety monitoring board. Partially, a clinical evaluation was also performed, focusing on the efficacy of the micro-graft with MGCT on peri-implant soft tissues regeneration.

### 2.5. Adverse Events (AEs)

All adverse events (AEs) that occurred between the surgery and 16 weeks after surgery were recorded. AEs are any undesirable experiences such as infection, pain, bleeding, swelling, fever, septic shock, and wound dehiscence. If necessary, the investigators administered appropriate treatments. A serious AE (SAE) was defined as any adverse reaction resulting in any of the following outcomes: a life-threatening condition or death; a condition that required inpatient hospitalization or prolongation of an existing hospitalization, threatening to cause disability or death; a congenital anomaly; or a birth defect. All SAEs were reported to the IRB and the certified committee for regenerative medicine by the responsible investigator.

### 2.6. Data Collection and Management

The personal data and the clinical data of the participants were coded and saved separately. All data were recorded in a case report form (CRF) by appropriate and authorized individuals (principal investigator or sub-investigators). All data related to personal information such as the consent form were kept in locked cabinets. The investigator maintained a personal identification list (patient numbers with the corresponding patient names) to enable records to be identified. The monitoring of study compliance and data collection was done by the principal investigator, sub-principal investigators, and authorized (not involved in this study) staff. During the study, the monitor made regular site visits to review protocol compliance, assess the laboratory procedures, and ensure that the study was being conducted according to protocol requirements.

### 2.7. Trial Status

The first version of research proposal was registered on 4 January 2017 and was last updated on 11 May 2020. Recruitment started in March 2017 and finished in July 2021.

### 2.8. Trial Registration

This study was registered with the Japan Registry of Clinical Trials (http://jrct.niph.go.jp) as jRCTs072180075, accessed on 26 March 2019 and the University Hospital Medical Information Network (trial no. UMIN000027150).

## 3. Results

Information about the study subjects is summarized in Table 1. Four patients (three females and one male) were enrolled in this clinical study, and the ages of the patients ranged from 58 to 71 years. According to postoperative blood tests results, none of the patients had any particular abnormalities.

Case 1. A 67-year-old woman who had inadequate keratinized tissue band in both mandibular molar regions (Figure 4) was submitted to a one-stage implant, with the implant placed on 36, 37, 46, and 47 and a micro-graft. A piece of gingival connective tissue was harvested from the palatal mucosa of both maxillary tuberosities. The collagen sheet (size 10 × 20 mm^2^) with MGCTs covered the periosteum surrounding the implants. Four weeks after the surgery, there was an expanded region of peri-implant mucosa at the buccal side, and this area was maintained 2 months after transplantation. Throughout the entire period, no AEs were observed.

Case 2. A 58-year-old woman who had less peri-implant mucosa in the left maxillary premolar region (Figure 5) with the implants placed on 22, 24, and 25 was submitted to a second-stage surgery at 5 months post-first surgery. The gingival connective tissue was harvested from the palatal mucosa of the left maxillary tuberosity, and a collagen sheet (size 5 × 20 mm^2^) was seated around the healing abutments. The reconstructed peri-implant mucosa was found surrounding the healing abutment at 4 weeks after surgery, and this region remained at 16 weeks. During the study protocol, no AEs were confirmed.

Case 3. A 71-year-old woman who had the implants placed on 25 and 26 was submitted to second-stage surgery at 6 months post-first surgery (Figure 6). There was little keratinized tissue band in the left maxillary premolar region. The non-keratinized tissue band was shifted to both buccal and palatal sides with split-thickness. From the palatal mucosa of the left maxillary tuberosity, the gingival connective tissue was harvested, and a collagen sheet (size 20 × 20 mm^2^) was seated around the healing abutments. Four weeks after surgery, the peri-implant mucosa was obviously increased surrounding the healing abutment. Throughout the whole period, no AEs were observed.

Case 4. A 68-year-old man who had the implants placed on 32 and 42 was submitted to second-stage surgery at 7 months post-first surgery (Figure 7). The keratinized tissue band was observed at the alveolar crest line. The gingival connective tissue was harvested from the palatal mucosa of the right maxillary tuberosity, and a collagen sheet (size 10 × 20 mm^2^) was seated at the labial side of the healing abutments (32 and 42). The reconstructed peri-implant mucosa was found on the labial side at 2 weeks after surgery. Four weeks after surgery, the peri-implant mucosa was observed around the healing abutments. Throughout the entire period, no AEs were observed.

## 4. Discussion

In this clinical study, we harvested a piece (approximately 3 mm^3^) of gingival connective tissue from the maxillary tuberosity. This procedure for harvesting the graft tissue has advantages to avoid a high level of skill and reduce the donor-site morbidity. Then, MGCTs provided sufficient width (5–20 × 20 mm^2^) and thickness of the peri-implant mucosa at the recipient site. Therefore, micro-grafts with MGCTs show the potential usefulness for keratinized tissue reconstruction and augmentation as an alternative treatment of FGG or CTG.

This is the first clinical study to report the safety and preliminary efficacy of micro-grafts with MGCTs for peri-implant mucosa regeneration in dental implant therapy. The first concern of this study was the safety of the micro-graft. No AEs occurred, including systemic complications, excessive immune reactions, severe infection, or excessive postoperative complications in the 16-week follow-up period. In addition, there was no peri-implant mucositis, and the stability of the dental implants was maintained after micro-graft transplantation. Therefore, this clinical study demonstrated that transplantation of micro-grafts with MGCTs could be performed safely for peri-implant mucosa reconstruction and showed no adverse effects in dental implant treatment.

As mentioned above, autologous MGCTs obtained by the tissue disruptor technique, which is a novel strategy for tissue mechanical disaggregation, were used. The technique allows autologous micro-grafts enriched with stem cells, such as MSCs, to be obtained during surgery and be ready for use without any cell manipulation process. Additionally, it has been claimed that the mechanical disaggregation is faster and safer than enzymatic techniques [36]. Therefore, the system has already been used in various fields of medicine and dentistry and has shown clinical efficacy [31,37,38]. The obtained micro-graft can be applied alone or in combination with different biological scaffolds, such as collagen. In the present study, the transplanted micro-graft was prepared by seeding the suspended autologous MGCTs using the tissue disruptor technique onto TERUDERMIS^®^, which has already been widely used in clinical practice [39,40,41]. The atelocollagen-matrix consists of a bottom layer, which is made of calf dermal collagen in which the telopeptide is eliminated by protease treatment, and a silicone top layer that prevents wound infection. Then, the TERUDERMIS^®^ collagen layer reconstructs dermis-like tissue by the patient’s own cells and capillary infiltration from contacted tissues. Therefore, due to the fact that both the tissue disruptor technique and atelocollagen-matrix have each been shown to be safe in clinical use, it was expected that the micro-graft could also be used safely in peri-implant mucosa reconstruction in humans. As expected, no AEs were identified during the observation periods.

Additionally, an assessment of the preliminary efficacy of micro-grafts on peri-implant mucosa regeneration in dental implant treatment was partially performed. The present clinical study showed that the transplanted micro-graft supported peri-implant mucosa augmentation in all cases. We previously reported that micro-grafts of porcine MGCTs using the tissue disruptor technique promoted skin wound healing in immune-deficient mice, and this strategy can contribute to peri-implant mucosa construction or augmentation [34]. This study also showed that the gingival mucosa-derived mesenchymal stem cells (GMMSCs) clearly induced vascularization during re-epithelialization in a paracrine manner. Interestingly, compared to dermal MSCs, GMMSCs have greater proliferation and migration capacity, and they promote wound healing [17,18,19,20]. Furthermore, GMMSCs possessed not only multipotent differentiation capacity, but also immunosuppressive and anti-inflammatory functions [18,20,23,42,43]. In regard to the paracrine functions of MGCTs including vascularization, our previous study showed increased expression levels of proangiogenic factors, such as ANG, EGF, FGF, IGF, IP-10, KC, MMP-3, -9, SDF-1, and VEGF, in MGCTs using a proteome profile array [34]. Therefore, gingival tissue enriched with growth factor, as well as stem/progenitor cells such as GMSCs, could be an excellent source for soft tissue engineering. In the present study, the efficacy of micro-grafts for peri-implant mucosa regeneration in dental implant therapy was confirmed in all cases. Notably, in the third case, there was little peri-implant mucosa around the dental implants. However, after transplantation of the micro-graft with MGCTs, reconstructed peri-implant mucosa was obviously found surrounding the healing abutment 4 weeks after surgery. Due to the lack of controlled clinical study, it was not possible to assess how much MGCTs contributed to the regeneration of peri-implant mucosa. Nevertheless, we consider that sufficient cells and growth factors might be supplied from transplanted MGCTs for peri-implant mucosa augmentation around the dental implants.

In this study, the included subjects were limited to four patients. Though the included subjects were limited to four patients for a first-in-human pilot study, the results of the present study demonstrated that using MGCTs did not cause any AEs. Meanwhile, this pilot case series is restricted to the safety assessment of the technique in solely four patients and more studies are necessary for the understanding of this micro-graft with MGCTs. Therefore, the preliminary outcome of this study enables us to proceed to the next randomized controlled trial to assess the efficacy of the micro-graft with MGCTs including more patients, clinical assessment regarding health of peri-implant soft tissue, and longer follow up.

## 5. Conclusions

In conclusion, although there were only four patients in this pilot case series, the results demonstrated that using micro-grafts with MGCTs did not cause any irreversible AEs, and they showed the efficacy of micro-grafts with MGCTs for peri-implant mucosa regeneration in dental implant therapy. Further studies, such as randomized controlled trials (RCT), are required to confirm both the safety and efficacy of micro-grafts, because we could not evaluate the efficiency of the micro-graft with MGCTs compared with that of atelocollagen-matrix transplantation. However, this strategy can be expected to have low invasiveness and be applied easily and safely in the clinic as an alternative to FGG or CTG.

## Figures and Tables

**Figure 1 medicines-10-00009-f001:**
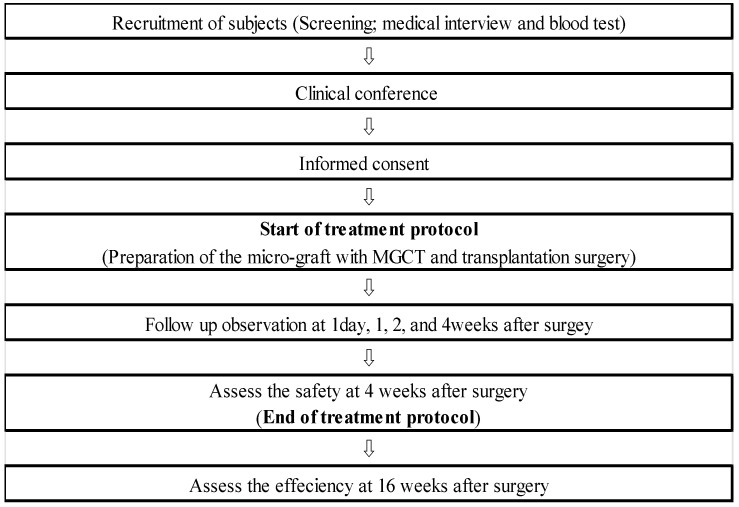
The procedure of pilot study. All processes of micro-graft with MGCTs preparation and transplantation surgery were performed in a clean operation at the Nagasaki University hospital. The safety verification was evaluated in terms of the clinical course until 4 weeks after surgery.

**Figure 2 medicines-10-00009-f002:**
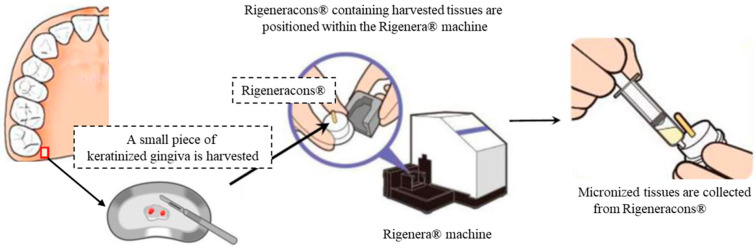
Procedures of preparing the micronized tissues using Rigenera^®^ system.

**Figure 3 medicines-10-00009-f003:**
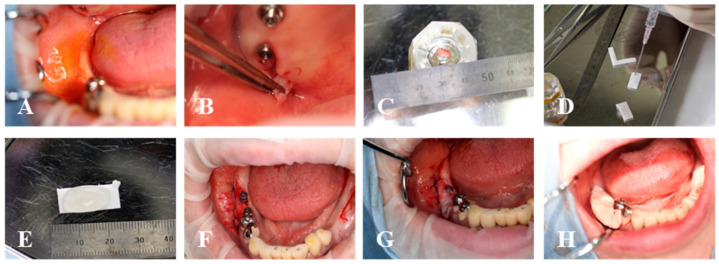
Procedures of micro-graft. (**A**); Iodine staining detects the keratinized tissue band area at the right molar region of the mandible. (**B**,**C**); A small piece of the gingiva is harvested from the maxillary tuberosity. (**D**,**E**); A piece of tissue is micronized by the tissue disruptor technique and the solution with micronized-gingiva is dripped onto the collagen sheet. (**F**); The peri-implant mucosa surrounding the implants are shifted apically with remaining the periosteum. (**G**); The exposed periosteum is covered by the collagen sheet with micronized-tissue. (**H**); The region was protected with a periodontal dressing after surgery.

**Figure 4 medicines-10-00009-f004:**
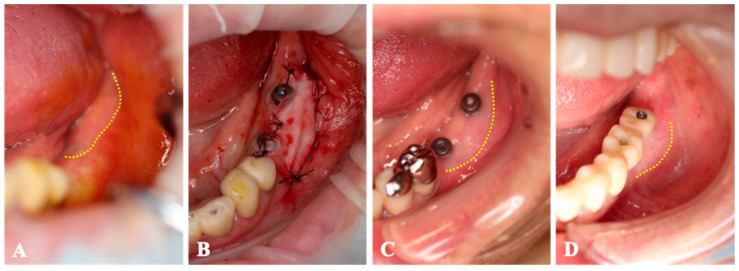
Case 1. (**A**); The keratinized tissue band was located at the alveolar crest (dotted-line). (**B**); The collagen sheet with micronized-gingiva covers the periosteum surrounding the implants. (**C**); The region of peri-implant mucosa is expanded at the buccal side (dotted-line) 4 weeks after transplantation. (**D**); The peri-implant mucosa region maintains its area 2 months after surgery.

**Figure 5 medicines-10-00009-f005:**
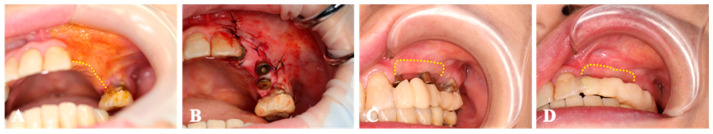
Case 2. (**A**); The keratinized tissue band is located at the alveolar crest (dotted-line). (**B**); The collagen sheet with micronized-gingiva is seated around the healing abutments. (**C**); The peri-implant mucosa is observed (dotted-line) 4 weeks after surgery. (**D**); The area of peri-implant mucosa is maintained (dotted-line) 16 weeks after surgery.

**Figure 6 medicines-10-00009-f006:**
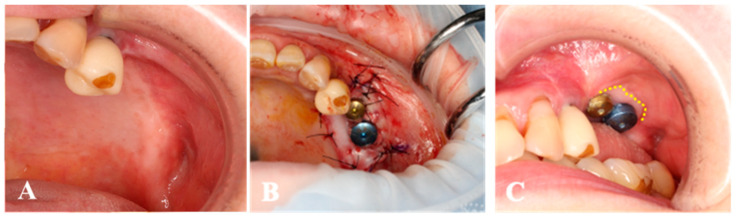
Case 3. (**A**); The keratinized tissue band is almost disappeared. (**B**); Non-keratinized tissue band is shifted both buccal and palatal sides with split-thickness. The collagen sheet with micronized-gingiva is seated around the healing abutments. (**C**); The peri-implant mucosa is found surrounding the healing abutment (dotted-line) 4 weeks after surgery.

**Figure 7 medicines-10-00009-f007:**
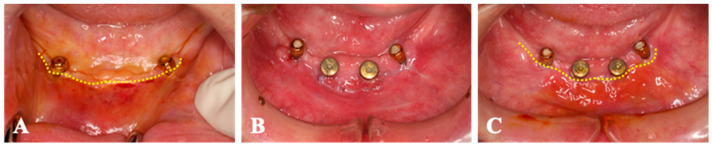
Case 4. (**A**); The keratinized tissue band is located at the alveolar crest (dotted-line). (**B**); The keratinized tissue band is shifted to the labial side 2 weeks after surgery. (**C**); The peri-implant mucosa is observed around the healing abutments (dotted-line) 4 weeks after surgery.

**Table 1 medicines-10-00009-t001:** List of the patients who participated in the clinical study.

Case	Gender	Age	Surgical Procedure	Implant Placement Site	Keratinized-Gingival Connective Tissue Harvested Site	TERUDERMIS^®^ Size
1	F	67	one-stage implant	76┬67	both palatal mucosa of maxillary tuberosity	10 × 20 mm
2	F	58	second-stage surgery (two-stage implant)	└245	left palatal mucosa of maxillary tuberosity	5 × 20 mm
3	F	71	second-stage surgery (two-stage implant)	└56	left palatal mucosa of maxillary tuberosity	20 × 20 mm
4	M	68	second-stage surgery (two-stage implant)	2┬2	right palatal mucosa of maxillary tuberosity	10 × 20 mm

## Data Availability

The datasets are available upon reasonable request to the corresponding author.

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
