# Peer review of "First-in-Human Study to Investigate the Safety Assessment of Peri-Implant Soft Tissue Regeneration with Micronized-Gingival Connective Tissue: A Pilot Case Series Study"

_medicines, 2023, doi:10.3390/medicines10010009_

Round 1

Reviewer 1 Report

I appreciate the originality of the study. 

- Why you applied the matrix as a FGG and not in a bilaminar tecnique?

- To access the efficacy of the matrix why you didn't make some linear or volumetric measurements? 

Author Response

-I appreciate the originality of the study. 

First of all, we are most appreciative of your time and constructive criticisms. The comments have helped us significantly improve our manuscript. We have addressed your comments with point-by-point responses. Please check our responses listed below.

- Why you applied the matrix as a FGG and not in a bilaminar technique?

We thank the reviewer for this excellent suggestion. The application to bilaminar technique (CTG) is very useful. However, the success of treatment in a CTG needs sophisticated technique compared with a FGG. The aim of this first-in-human study was to assess the safety of micro-grafts with MGCTs with a small number of patients before we proceed to the next large-scale clinical trial. Therefore, regardless of operator’s technique, a FGG was suitable technique for the evaluation of the safety. In the next phase, as you proposed, we will evaluate efficacy of using a FGG and CTG.

- To access the efficacy of the matrix why you didn't make some linear or volumetric measurements?

We appreciate the reviewer for the appropriate comment. As mentioned above, in this first-in-human study, we evaluated the safety of the micro-grafts with MGCTs with a small number of patients. We would like to report the safety of the micro-grafts with MGCTs with a small-scale study, before proceeding to the next large-scale clinical study to assess the efficacy. The aim of this clinical study was just to assess the safety of the micro-graft with MGCTs. We put less emphasis on the evaluation of the efficacy and, therefore, just observed to confirm whether the adherent gingiva could be reconstructed without any problems. This study had neither a control group nor a group in which the atelocollagen sheet without MGCTs was just transplanted, and did not include an accurate evaluation of the effectiveness. It was conducted to confirm that the protocol could be implemented as a treatment without any troubles in a small number of cases. Therefore, in next clinical study, we will assess the efficacy of the micro-grafts with MGCTs for the gingiva reconstruction not only as a FGG but also a CTG. We would really appreciate it if you understand the aim of this study and our perspectives.

Reviewer 2 Report

The manuscript is well written but some comments need to be addressed;

Comments of abstract

1-     Line 14 : remove “The” before “safety” .

2-     Line 24: add “a” before “high”.

Comments of introductions

1-     Paragraph in line 49 :“ meanwhile……. mechanisms [14, 15]”required rephrasing to clear the information.

2-     Line 57: Add “ the” before “treatment of gingival defects”.

3-     Line 88: add “ a/the” before “potential”.

Comments of material and methods

1-     More details should be mentioned in section of “2.2 Preparation of micro-graft with micronized- gingival connective tissue”, how the device works, the adjustment of the device, the shape and features of the micrograft, the handling of mico-graft by instruments before grafting, instruments used in the surgery and type of suture and suturing technique.

2-     Figure 2 should be focused on the working sites with more detailed steps.

3-     Figure 5 A: not clear.

Comments of discussion

1-     Line 301: replace “ period” by “ periods” .

2-     Line329: replace “perspective” by “ perspectives”.

Author Response

-The manuscript is well written but some comments need to be addressed;

Thank you for giving us the opportunity to strengthen our manuscript with your valuable comments and queries. We have worked hard to incorporate your feedback and hope that these revisions persuade you to accept our submission.

Comments of abstract

1-     Line 14 : remove “The” before “safety” .

2-     Line 24: add “a” before “high”.

According to your kind suggestion, we revised the Abstract section (highlighted sentences by yellow in Line 15 and 25) as follows.

              (In line 15) “However, safety and efficacy have not yet been established in patients.”

              (In line 25) “~without requiring a high level of skills and morbidity to harvest graft tissues”

 Comments of introductions

1-     Paragraph in line 49:“ meanwhile……. mechanisms [14, 15]”required rephrasing to clear the information.

2-     Line 57: Add “the” before “treatment of gingival defects”.

3-     Line 88: add “a/the” before “potential”.

We appreciate the reviewer for the appropriate comment. Following your suggestion, we revised the Introduction section (highlighted sentences by yellow in Line 47, 56 and 98) as follows.

              (In line 47) “Meanwhile, mesenchymal stromal cells (MSCs) have identified numerous mechanisms by which these cells can attenuate the acute inflammatory response, and promote re-epithelialization and extracellular matrix remodeling via their paracrine mechanisms.

              (In line 56) “These properties have endowed GMMSCs with potent regenerative and therapeutic potentials in the treatment of gingival defects.”

              (In line 98) “The micro-graft with MGCTs has a potential as an alternative therapy to the conventional FGG or CTG in the clinical setting because of its low morbidity and ready availability.”

 Comments of material and methods

  • More details should be mentioned in section of “2.2 Preparation of micro-graft with micronized- gingival connective tissue”, how the device works, the adjustment of the device, the shape and features of the micrograft, the handling of mico-graft by instruments before grafting, instruments used in the surgery and type of suture and suturing technique.

Thank you for your comments, which help us to greatly improve the manuscript. We agree with you and have incorporated this suggestion in Material and Methods. For these additional descriptions, our manuscript is revised in the Material and Methods section (highlighted sentences by yellow) as follows.

              (In line 154) “With 1 ml of saline solution per piece, the harvested tissue was micronized for 90 seconds by the Rigenera® machine. In using the device, all processes are fully automatic. Then, micronized-gingival connective tissues (MGCTs, ≤100 μm each) were suspended in the saline and collected into a 1.5-ml sterilized tube. Subsequently, the suspended micronized-gingival connective tissue was seeded onto a square shaped atelocollagen-matrix (TERUDERMIS®; Olympus Terumo Biomaterials, Tokyo, Japan), which was put with the silicone membrane face-down on the sterilized dish just before the transplantation. An atelocollagen-matrix was adjusted for suitable size of the exposed periosteum by handling with straight tweezers and surgical scissors before seeding.

              (In line 171)” using non-absorbable thread (Nylon) with locked single-layer suturing techniques. After applying it to the wound, a periodontal dressing was placed."

  • Figure 2 should be focused on the working sites with more detailed steps.

We appreciate your comments. Following your advice, more comprehensible images of working sites (Fig. 3D) have been added in revised Figure 3 (Table1 has been termed as Figure 1. Therefore, the Figure 2 has been termed as Figure 3). We hope the revised figures respond to your suggestions.

3-     Figure 5 A: not clear.

 According to your suggestion, we have also revised Figure 6 (Table1 has been termed as Figure 1. Therefore, the Figure 5 has been termed as Figure 6). We hope that these revisions will be satisfactory.

Comments of discussion

We appreciate the reviewer for the appropriate comment. Following your suggestion, we revised the Discussion section (highlighted sentences by yellow in line 316) as follows.

  • Line 301: replace “period” by “periods”.

              (In line 316) “As expected, no AEs were identified during the observation periods.”

  • Line329: replace “perspective” by “perspectives”.

We are deeply apologized for not accomplish to delete the annotation in the template file downloaded from medicines. The sentence including “perspective” was a part of unnecessary annotation, which should be deleted. We sincerely apologize again for your confusion and have cleared these annotative sentences completely.

Reviewer 3 Report

Comments to authors

Dear authors,

Thank you for submitting the manuscript entitled "Clinical Safety Assessment of Keratinized Gingival Tissue Regeneration" for potential publication in Medicines journal. This article addresses an interesting topic for the dental community, and fit to the scope of the journal. Overall, the manuscript is well organized and coherent. However, I consider that several important clarifications are necessary from the part of the authors before publication. You can find below my comments.

Title:

- The title of this manuscript does not describe appropriately the content of the paper. First, keratinized gingival tissue is a pleonasm since the gingival tissue is keratinized. Furthermore, “clinical safety assessment” can be confusing to readers. Please, clarify to readers in the title of the manuscript what they might expect when reading “clinical safety assessment.” Finally, it is important to mention in the title that this is a pilot case series. People may feel misleading when deepen into the paper and discover that it is not a “complete” study, but a pilot one.

Abstract:

- An important concept should be corrected by the authors along the whole manuscript: there is no gingival tissue around dental implants! What we have is a peri-implant soft tissue, which it is not the same as gingival tissue. Therefore, I kindly ask the authors for revising the paper accordingly.

- Please revise the keywords. Avoid using acronyms and abbreviations. Moreover, try to use MeSH terms related to the methods employed or the study design as keywords.

Introduction:

- Generally speaking, the article is not well written. There are many sentences of grammatical concern. I.e., line 45, “… have been showed…” and line 52 “… have been shown…” The authors should improve the English language use. Therefore, I recommend the authors to proof-read the whole manuscript performing these corrections. A professional English service is recommended.

- The Introduction section is nice and informative. However, the affirmations were not well support by the references in my opinion. Please, check the references used in the manuscript and bring more data of comparisons made in those original studies that justify your arguments.

- I believe the authors should avoid the use of commercial name of materials or keep it to a minimum. Even though the author declare that do not have conflicts of interest, this paper seems like a commercially-driven publication. In my opinion, the authors should discuss the data of techniques and materials with minimal mention to brands.

- Despite very interesting, the clinical use of MGCT is not very well documented in preclinical studies, right? Are there evaluations of the safety in preclinical (animal) models? It seems to me that MGCTs need to be further verified by more in vivo and in vitro experiments.

Materials and Methods:

- The manuscript needs special attention in this section. Important information is missing.

- What was the method employed for evaluating your primary endpoint? The authors limited their arguments on “The primary endpoint was safety/tolerability”, but this information should be followed by the method employed for evaluation of safety or tolerability.

- English revision required. For example, line 108: underwent = were submitted. Please revise the whole manuscript in this matter.

- Table 1 is not a table. Actually, it is a flowchart, and should be mentioned as a Figure, note Table. Moreover, it would be interesting if the authors disclose the number of individuals that were screened and how many people were excluded from the study, with the reasons for exclusion.

- Table 1: “infromed” = informed. “Start of protocol treatment” = Start of treatment protocol. Please revise the whole text.

- Please, use bullet points instead of numbers for the inclusion and exclusion criteria.

- Please, inform the centrifugation protocol for the micronized tissue.

- After reading the surgical procedure, I understood the intention of the authors on the proposed method. Although interesting, it does not mean much to put particles of soft tissue harvested from the palate onto a collagen matrix in a de-epithelized peri-implant mucosa. We cannot trust that the micronized-graft is really producing a clinically relevant effect. This principle should be tested in preclinical models first. With some histological results from animal experiments, maybe the authors can argue that the proposed intervention is efficient in humans. Moreover, a comparative study with collagen matrix without micronized graft is required to confirm this principle in patients. To summarize, this study needs more experiments before publication.

- “Wound irrigation was performed 1 day and 1 week after the surgery”. What was the solution used for irrigation?

- “The safety assessment was conducted by an independent safety monitoring board.” What was the criteria used for this safety evaluation? Which parameters did you assessed? Please clarify to readers.

- “a clinical evaluation was also performed, focusing on the efficacy of the micro-graft with MGCT on keratinized gingival tissue regeneration.” What were the parameters observed and how did you measure them?

- Very few data can be extracted from the proposed experiments. I regret to say that, in my opinion, this investigation cannot be published in the current format. More experiments must be performed before publication.

Results:

- Four patients are not enough to confirm the proposed outcomes. My vision is that the authors need to include more patients in the study and clearly define the analyzed parameters.

- Basically, what the authors presented in the Results section was a case series.

Discussion:

- The adverse events evaluation should ideally be performed in hundreds of patients. We cannot believe that an analysis of four cases is evidence for safety of a clinical procedure.

Conclusion:

- Results were so preliminary that the authors cannot end up with any solid conclusion. Please, consider the suggestions mentioned above and resubmit the manuscript after convenient modifications.

Author Response

-Dear authors,

Thank you for submitting the manuscript entitled "Clinical Safety Assessment of Keratinized Gingival Tissue Regeneration" for potential publication in Medicines journal. This article addresses an interesting topic for the dental community, and fit to the scope of the journal. Overall, the manuscript is well organized and coherent. However, I consider that several important clarifications are necessary from the part of the authors before publication. You can find below my comments.

We are most appreciative of your constructive comments on our manuscript (medicines-2069442). We hereby submit a revised version of our manuscript re-entitled “First-in-human study to investigate the safety assessment of peri-implant soft tissues regeneration with micronized-gingival connective tissues: A Pilot Case Series Study”. Please note that the title has been modified responding to your suggestion. All comments from you have been addressed. To the best of our ability, we have tried to answer and comply with requested changes. We are grateful for your consideration in reviewing our revised manuscript and are looking forward to your response.

              (In line 2) “First-in-human study to investigate the safety assessment of peri-implant soft tissues regeneration with micronized-gingival connective tissues: A Pilot Case Series Study

Title: Clinical Safety Assessment of Gingival Tissue Regeneration

- The title of this manuscript does not describe appropriately the content of the paper. First, keratinized gingival tissue is a pleonasm since the gingival tissue is keratinized. Furthermore, “clinical safety assessment” can be confusing to readers. Please, clarify to readers in the title of the manuscript what they might expect when reading “clinical safety assessment.” Finally, it is important to mention in the title that this is a pilot case series. People may feel misleading when deepen into the paper and discover that it is not a “complete” study, but a pilot one.

We agree with you and have incorporated this meaningful suggestion. To describe the content of our manuscript, we have corrected the title, “First-in-human study to investigate the safety assessment of peri-implant soft tissues regeneration with micronized-gingival connective tissues: A Pilot Case Series Study”. Additionally, we have replaced the term “keratinized gingival tissue” throughout the paper with “peri-implant soft tissues” to use more precise terms.

Abstract:

- An important concept should be corrected by the authors along the whole manuscript: there is no gingival tissue around dental implants! What we have is a peri-implant soft tissue, which it is not the same as gingival tissue. Therefore, I kindly ask the authors for revising the paper accordingly.

We agree with you and have incorporated this suggestion. As described previously, we have replaced the term “keratinized gingival tissue” throughout the paper with “peri-implant soft tissues” to use more precise terms.

- Please revise the keywords. Avoid using acronyms and abbreviations. Moreover, try to use MeSH terms related to the methods employed or the study design as keywords.

 We are grateful for the helpful suggestions. Following your useful advice, we have now revised the keywords (highlighted by yellow in line 26) using MeSH terms as follows.

              (In line 26) Keywords: gingiva; connective tissues; clinical study; regeneration

Introduction:

- Generally speaking, the article is not well written. There are many sentences of grammatical concern. I.e., line 45, “… have been showed…” and line 52 “… have been shown…” The authors should improve the English language use. Therefore, I recommend the authors to proof-read the whole manuscript performing these corrections. A professional English service is recommended.

Following Reviewer’s advice, our manuscript has been corrected by a professional English service and our colleagues whose native language is English. Attached is certificate confirms from a professional English service. We hope our revised manuscript has been improved enough for readers to understand.

              (In line 45) “ADM and CM have shown utility in augmenting…”

              (In line 51)” …, mesenchymal stem cells (GMMSCs) have shown to promote wound healing…”

- The Introduction section is nice and informative. However, the affirmations were not well support by the references in my opinion. Please, check the references used in the manuscript and bring more data of comparisons made in those original studies that justify your arguments.

Thank you for your comments, which help us to greatly improve the manuscript. We have provided the additional data of comparisons made in those original studies that justify our arguments in the Introduction section.

              (In line 82) “In fact, mesenchymal cells derived from MGCTs were detected mainly in the matrix immediately below the epithelialization area. The expression of mRNAs related to endothelial and epithelial cell marker genes (cd31, vWF, cytokeratin, vegf-a, vegf-b, flt-1, and flk-1) were elevated in transplanted samples. Additionally, our data using proteome profile array showed the increased expression levels of proangiogenic factors, such as Angiogenin, EGF, FGF, IGF, IP-10, KC, MMP-3, −9, SDF-1, and VEGF, in MGCTs. Therefore, surviving mesenchymal cells derived from MGCTs promoted vascularization and re-epithelialization in a paracrine manner. Moreover, while a small number of MGCT derived cells may have contributed directly to form the blood vessels or epidermis in the wound area, the majority of donor cells likely functioned as mesenchymal cells. Then, the advantage of this strategy is that the MGCTs from a small piece (approximately 3 mm square) of maxillary tuberosity can regenerate the skin wound approximately 20-fold as large as harvested tissue. Our findings suggested that this strategy could easily be applied to the clinical setting, showing low morbidity and ready availability, and it could contribute to keratinized gingival tissue reconstruction or augmentation using the Rigenera® system.”

- I believe the authors should avoid the use of commercial name of materials or keep it to a minimum. Even though the author declare that do not have conflicts of interest, this paper seems like a commercially-driven publication. In my opinion, the authors should discuss the data of techniques and materials with minimal mention to brands.

Following your advice, we revised our manuscript to keep the use of commercial name of materials to a minimum throughout the paper.

- Despite very interesting, the clinical use of MGCT is not very well documented in preclinical studies, right? Are there evaluations of the safety in preclinical (animal) models? It seems to me that MGCTs need to be further verified by more in vivo and in vitro experiments.

We thank the reviewer for this comment. The MGCTs consist of autologous micronized-gingival connective tissues and other groups also reports the optimal regenerative potential of micronized tissues obtained from various human samples, such as periosteum, dental pulp, adipose, or dermal tissues, displayed high cell viability, and optimal regenerative potential (Giaccone et al., 2014; Trovato et al., 2015; Marcarelli et al., 2017; Monti et al., 2017). Indeed, Svolacchia et al. demonstrated that micro-graft of skin (dermal) connective tissues showed effectiveness in improving surgical wound dehiscence or exaggerated scars in clinical trials (Svolacchia et al., 2016). These data suggested the MGCTs have a high safety, however, the safety of them have not yet been established in patients. Therefore, we performed this clinical study to assess the safety and the preliminary efficacy of micro-grafts with MGCTs in a pilot study.

Materials and Methods:

- The manuscript needs special attention in this section. Important information is missing.

- What was the method employed for evaluating your primary endpoint? The authors limited their arguments on “The primary endpoint was safety/tolerability”, but this information should be followed by the method employed for evaluation of safety or tolerability.

We apologize for not providing readers with enough information. Following Reviewer’s suggestions, we have revised our manuscript in Materials and Methods section (highlighted sentences by yellow in line 189 and 196) as follows.

              (In line 189) “The primary endpoint was the safety/tolerability of the protocol. The safety and tolerability were assessed by the presence or the absence of an adverse events (AEs). The safety assessment was conducted by an independent safety monitoring board.”

              (In line 196) “AEs are any undesirable experiences such as infection, pain, bleeding, swelling, fever, septic shock, and wound dehiscence. If necessary, the investigators administered appropriate treatments.”

- English revision required. For example, line 108: underwent = were submitted. Please revise the whole manuscript in this matter.

As above, our manuscript has been corrected by a professional English service and our colleagues whose native language is English.

              (In line 118) “The participants were enrolled according to the following inclusion criteria and interviewed regarding their underlying health conditions and blood tests before the operation.”

- Table 1 is not a table. Actually, it is a flowchart, and should be mentioned as a Figure, note Table. Moreover, it would be interesting if the authors disclose the number of individuals that were screened and how many people were excluded from the study, with the reasons for exclusion.

We appreciate the appropriate comments. Table 1 has been mentioned as Figure 1. Indeed, four people were screened and no ones were excluded from this study. Then four patients were enrolled to this study as described in manuscript. This first-in-human study was designed to assess the safety and the preliminary efficacy of micro-grafts with MGCTs in a pilot study to proceed to the next step for clinical use.

- Table 1: “infromed” = informed. “Start of protocol treatment” = Start of treatment protocol. Please revise the whole text.

We apologize again for the inconvenience that I have caused you. We have revised Figure 1 as follows.

                            (Figure 1) “Informed consent

                            (Figure 1) “Start of treatment protocol

- Please, use bullet points instead of numbers for the inclusion and exclusion criteria.

Following Reviewer’s suggestions, our manuscript has been revised (highlighted sentences by yellow in line 123 and 131) as follows.

              (In line 123) “Patient Inclusion Criteria

              (In line 131) “Patient Exclusion Criteria

- Please, inform the centrifugation protocol for the micronized tissue.

We apologize for not explaining sufficiently. In fact, the procedure to prepare the micronized tissues by the Rigenera® machine does not include the process of centrifugation. For additional explanation, our manuscript is revised in the Material and Methods section (highlighted sentences by yellow) as follows.

              (In line 154) “In using the device, all processes are fully automatic. Then, micronized-gingival connective tissues (MGCTs, ≤100 μm each) were suspended in the saline and collected into a 1.5-ml sterilized tube. Subsequently, the suspended micronized-gingival connective tissue was seeded onto a square shaped atelocollagen-matrix (TERUDERMIS®; Olympus Terumo Biomaterials, Tokyo, Japan), which was put with the silicone membrane face-down on the sterilized dish just before the transplantation. An atelocollagen-matrix was adjusted for suitable size of the exposed periosteum by handling with straight tweezers and surgical scissors before seeding.

- After reading the surgical procedure, I understood the intention of the authors on the proposed method. Although interesting, it does not mean much to put particles of soft tissue harvested from the palate onto a collagen matrix in a de-epithelized peri-implant mucosa. We cannot trust that the micronized-graft is really producing a clinically relevant effect. This principle should be tested in preclinical models first. With some histological results from animal experiments, maybe the authors can argue that the proposed intervention is efficient in humans. Moreover, a comparative study with collagen matrix without micronized graft is required to confirm this principle in patients. To summarize, this study needs more experiments before publication.

We appreciate the Reviewer for appropriate comment. As mentioned in our preclinical report (Noda et al., J Cell Physiol. 2018 Jan;233(1):249-258), micronized-gingival connective tissues (MGCTs) obtained from a few millimeters of gingiva, reliably hastened the re-epithelialization of wounds with less scar contraction. Our histological results from animal experiments also revealed that MGCTs obviously induced vascularization as explants during the re-epithelialization process, and surviving mesenchymal cells derived from MGCTs might have vascularization and re-epithelialization in a paracrine manner. We have provided the additional and detailed explanations in the Introduction section to justify our arguments. Meanwhile, we agree with you, further investigations using a comparative study with collagen matrix without micronized graft is required. However, the primary aim of this first-in-human study was to assess the safety of micro-grafts with MGCTs. Though the included subjects were limited to only 4 patients for a first-in-human pilot study, the results of the present study demonstrated that using MGCTs did not cause any AEs. Then, the outcome of this study enables us to proceed further investigations on humans in order to confirm the safety and the efficacy of MGCTs setting control group. We hope these additional descriptions respond to your suggestions.

              (In line 82) “In fact, mesenchymal cells derived from MGCTs were detected mainly in the matrix immediately below the epithelialization area. The expression of mRNAs related to endothelial and epithelial cell marker genes (cd31, vWF, cytokeratin, vegf-a, vegf-b, flt-1, and flk-1) were elevated in transplanted samples. Additionally, our data using proteome profile array showed the increased expression levels of proangiogenic factors, such as Angiogenin, EGF, FGF, IGF, IP-10, KC, MMP-3, −9, SDF-1, and VEGF, in MGCTs. Therefore, surviving mesenchymal cells derived from MGCTs promoted vascularization and re-epithelialization in a paracrine manner. Moreover, while a small number of MGCT derived cells may have contributed directly to form the blood vessels or epidermis in the wound area, the majority of donor cells likely functioned as mesenchymal cells. Then, the advantage of this strategy is that the MGCTs from a small piece (approximately 3 mm square) of maxillary tuberosity can regenerate the skin wound approximately 20-fold as large as harvested tissue. Our findings suggested that this strategy could easily be applied to the clinical setting, showing low morbidity and ready availability, and it could contribute to keratinized gingival tissue reconstruction or augmentation using the Rigenera® system.”

- “Wound irrigation was performed 1 day and 1 week after the surgery”. What was the solution used for irrigation?

              (In line 183) “Wound irrigation was performed with saline 1 day and 1 week after the surgery, …”

- “The safety assessment was conducted by an independent safety monitoring board.” What was the criteria used for this safety evaluation? Which parameters did you assessed? Please clarify to readers.

As mentioned above, we have revised our manuscript (highlighted sentences by yellow in line 189 and 196) as follows.

              (In line 189) “The safety and tolerability were assessed by the presence or the absence of an adverse events (AEs). The safety assessment was conducted by an independent safety monitoring board.”

              (In line 196) “All adverse events (AEs) that occurred between the surgery and 16 weeks after surgery were recorded. AEs are any undesirable experiences such as infection, pain, bleeding, swelling, fever, septic shock, and wound dehiscence. If necessary, the investigators administered appropriate treatments.”

- “a clinical evaluation was also performed, focusing on the efficacy of the micro-graft with MGCT on keratinized gingival tissue regeneration.” What were the parameters observed and how did you measure them?

We appreciate the reviewer for the appropriate comment. Following a recommendation from the Clinical Research Review Board, we assessed the safety of micro-grafts with MGCTs in first-in-human with a small number of patients before we proceed to the next large-scale clinical trial, since this was the first attempt to use the Rigenera system. We are now proceeding to design large-scale clinical trial, which will include a tidy assessment of the efficacy such as measuring the thickness of the gingiva with a probe. We would really appreciate it if you understand the aim of this study and our perspectives.

- Very few data can be extracted from the proposed experiments. I regret to say that, in my opinion, this investigation cannot be published in the current format. More experiments must be performed before publication.

 As the reviewer has pointed out, this FIH study could not provide meaningful data regarding the efficacy of the micro-graft with MGCTs. However, in order to proceed to the next randomized controlled trial to evaluate the efficacy of the micro-graft with MGCTs, this FIH study was a prerequisite and meaningful step to demonstrate that this treatment strategy could be conducted safely. This new treatment was performed based on a simple protocol that a small amount of gingival tissue was harvested, minced, seeded in a collagen matrix, and transplanted. Therefore, it was a strategy that was unlikely to cause serious adverse events in terms of safety. However, on the other hand, this new strategy contains an unapproved technique. Therefore, the Clinical Research Review Board requested us to first confirm that the treatment protocol can be implemented smoothly and safely, even in a small number of cases, and after publication, to proceed to the next randomized controlled trial to evaluate the efficacy of the micro-graft with MGCTs. The safety assessment in this FIH study that this treatment protocol could be performed without harming patients should be disclosed to public. We would appreciate it if you understand the importance of the phased implementation and publication of clinical study.

Results:

- Four patients are not enough to confirm the proposed outcomes. My vision is that the authors need to include more patients in the study and clearly define the analyzed parameters.

Thank you so much for your meaningful criticisms. Above all, this protocol consists of collecting a small amount of autologous tissue and transplanting it together with atelocollagen-matrix that have been clinically applied, so there are no safety concerns. However, even though it is a small amount (approximately 3 mm3), we could not deny a possibility that these invasions and the transplantation brought unexpected AEs such as pain, and wound dehiscence. Therefore, we needed to assess the safety of micro-grafts with MGCTs in first-in-human with a small number of patients. After publication of the safety of using this micro-graft, we will swiftly proceed to the next randomized controlled trial to assess the efficacy of the micro-graft with MGCTs on an appropriate scale. We hope for your kind understanding. Additionally, following your suggestions, we have revised our manuscript (highlighted sentences by yellow in line189, 196, and 342) as follows.

              (In line 189) “The safety and tolerability were assessed by the presence or the absence of an adverse events (AEs). The safety assessment was conducted by an independent safety monitoring board.”

              (In line 196) “AEs are any undesirable experiences such as infection, pain, bleeding, swelling, fever, septic shock, and wound dehiscence. If necessary, the investigators administered appropriate treatments.”

              (In line 342) In this study, the included subjects were limited to four patients. Though the included subjects were limited to four patients for a first-in-human pilot study, the results of the present study demonstrated that using MGCTs did not cause any AEs. Therefore, the outcome of this study enables us to proceed to the next randomized controlled trial to assess the efficacy of the micro-graft with MGCTs on an appropriate scale.

- Basically, what the authors presented in the Results section was a case series.

Although this treatment with micro-graft was considered to be highly safe, since it was a completely new treatment method, the presence or absence of the unexpected AEs such as pain, swelling, fever, and wound dehiscence and the effectiveness of micro-graft with MGCTs needed to be evaluated during the operative or post-operative course, even in a small number of cases. Therefore, by carefully making a commentary regarding each treatment and evaluation of therapeutic progress, we would like to demonstrate that this treatment protocol could be performed on schedule not only absence of AEs.

Discussion:

- The adverse events evaluation should ideally be performed in hundreds of patients. We cannot believe that an analysis of four cases is evidence for safety of a clinical procedure.

We thank you again for your meaningful criticisms. As aforementioned, this treatment with micro-graft was considered to be highly safe because this strategy needed much smaller volume of autologous tissue compared with FGG or CTG. However, the treatment was performed according to a new protocol as follows: (1) autologous tissue was harvested albeit a small amount; (2) MGCTs were obtained via Rigenera system; (3) the micro-graft was transplanted in combination with artificial material. Therefore, we planned to assess the safety of using micro-grafts with MGCTs with a small number of patients before proceeding to the randomized controlled trial on an appropriate scale. Then, first of all, we evaluated whether this treatment protocol could be performed on schedule or not in this first-in-human study with a small number of patients.

Conclusion:

- Results were so preliminary that the authors cannot end up with any solid conclusion. Please, consider the suggestions mentioned above and resubmit the manuscript after convenient modifications.

Based on our explanations and revises as described above, we would appreciate it if you could understand the validity of conducting this study in a small number of cases. We hope the revised version is now suitable for the publication and look forward to hearing from you.

Round 2

Reviewer 3 Report

Comments to authors

Dear authors,

Thank you for submitting the revised version of the manuscript entitled "First-in-human study to investigate the safety assessment of peri-implant soft tissues regeneration with micronized-gingival connective tissues: A Pilot Case Series Study" for potential publication in Medicines journal. The article improved a lot from the previous version and I congratulate the authors for their commitment during the review process. Despite, I consider that some points still need attention before paper publication. You can find below my comments.

Title:

- The title looks much better in the current version. Despite, I would change the word “tissues” for “tissue” in both mentions in the title.

Abstract:

- “We have recently developed an alternative strategy…” I would not say that the authors developed a distinguished strategy for treating peri-implant soft tissue deficiencies, because the technique has not been proven in human studies. Therefore, my suggestion is to say that the authors are proposing this new approach. In addition, the background section of the abstract seems too long. I recommend the authors to be concise and suppress unnecessary information. Please, rephrase sentences in the abstract.

- You should perform the modifications made in the rest of the manuscript in the abstract section. I missed the commentaries on the specific objectives, that this manuscript describes a case series, etc.

- Unnecessary mention to brands is present in the abstract. Please, consider removing commercial names from this section.

- Authors must include in the whole manuscript that this is a pilot case series.

Introduction:

- Even though the authors affirmed they performed a language revision in the whole manuscript, very few modifications were present and many English use mistakes can be seen. Please, proof-read the manuscript again and edit the text accordingly.

Materials and Methods:

- Legends from Figure 1 are missing.

- As requested in the last round of revision, please, use bullet points instead of numbers for the inclusion and exclusion criteria. The use of numbers can be confusing.

- I still can see in Figure 1 the term “protocol treatment” instead of “treatment protocol.” Care must be taken during the revision process. The authors should be conscious and aware to avoid repetition of mistakes. Please, revise the WHOLE manuscript to the next round of revision.

- Again, very few data can be extracted from the proposed experiments. Considering this, I would include in the Discussion section the limitations of this study. The authors might say that this pilot case series is restricted to the safety assessment of the technique in solely 4 patients, and that more studies are necessary for the understanding of this possible graft.

Results:

- Line 231: patients were submitted, not underwent. English revision is still necessary.

- From a personal standpoint, such a small sample size (4 patients), with such a limited analysis (only adverse events occurrence, which is very rare in Dentistry), with such a short evaluation time (4 weeks) cannot compose a full article publication. My vision is that the authors need to include more patients, include more parameters (maybe clinical assessment of peri-implant health, PD, BOP, KMW, etc.), and longer follow up (in four weeks not even the graft is fully recovered).

Discussion:

- I understand the authors and agree that your case series is the first step for the following RCT with the proposed method. However, we cannot trust in data provided from four cases and limited to a single safety assessment analysis. Moreover, the authors should clearly mention the limitations of the study in the Discussion section, as well as the recommendations for future investigations. 

Author Response

Dear authors,

Thank you for submitting the revised version of the manuscript entitled "First-in-human study to investigate the safety assessment of peri-implant soft tissues regeneration with micronized-gingival connective tissues: A Pilot Case Series Study" for potential publication in Medicines journal. The article improved a lot from the previous version and I congratulate the authors for their commitment during the review process. Despite, I consider that some points still need attention before paper publication. You can find below my comments.

We are most appreciative of your time, and thoughtful and constructive feedback you provided regarding our manuscript (medicines-2069442). We are pleased to submit a revised version of our manuscript entitled “First-in-human study to investigate the safety assessment of peri-implant soft tissue regeneration with micronized-gingival connective tissue: A Pilot Case Series Study.” To the best of our ability, we have tried to answer and comply with requested changes. We are grateful for your consideration in reviewing our revised manuscript and are looking forward to your response. We hope that these revisions are sufficient to make our manuscript suitable for publication in Medicines.

Title:

 - The title looks much better in the current version. Despite, I would change the word “tissues” for “tissue” in both mentions in the title.

We appreciate your appropriate comment. According to your kind suggestion, we revised the Title (In line 3 and 4) as follows.

              (In line 3 and 4) “First-in-human study to investigate the safety assessment of peri-implant soft tissue regeneration with micronized-gingival connective tissue: A Pilot Case Series Study.”

Abstract:

- “We have recently developed an alternative strategy…” I would not say that the authors developed a distinguished strategy for treating peri-implant soft tissue deficiencies, because the technique has not been proven in human studies. Therefore, my suggestion is to say that the authors are proposing this new approach. In addition, the background section of the abstract seems too long. I recommend the authors to be concise and suppress unnecessary information. Please, rephrase sentences in the abstract.

Thank you for your comments, which help us to greatly improve the manuscript. We have revised and rephrased in the Background section of the Abstract to exclude unnecessary information as follows.

              (In line 11) “We have recently proposed an alternative strategy of free gingival graft (FGG) and connective tissue graft (CTG)….”

              (In line 13) ”The advantage of this strategy is that MGCTs from a small piece (approximately 3mm square) of ….”

              (In line 14) “The advantage of this strategy is that MGCTs from a small piece of maxillary tuberosity can regenerate the gingival tissue approximately 20-fold as large as har-vested tissue.

              (In line 14) “This clinical study was a pilot case series, and the objective was to assess the safety and the preliminary efficacy of MGCTs on gingival tissue regeneration in dental implant therapy.”

- You should perform the modifications made in the rest of the manuscript in the abstract section. I missed the commentaries on the specific objectives, that this manuscript describes a case series, etc.

Thank you so much for your meaningful criticisms. Following your suggestions, we have revised Abstract section as follows.

              (In line 14) “This clinical study was a pilot case series, and the objective was to assess the safety and the preliminary efficacy of MGCTs on gingival tissue regeneration in dental implant therapy.”

              (In line 23) “Though further studies are needed on an appropriate scale, as an alternative strategy of FGG or CTG, MGCTs might be promising for gingival tissue reconstruction without requiring a high level of skills and morbidity to harvest graft tissues.”

- Unnecessary mention to brands is present in the abstract. Please, consider removing commercial names from this section.

We agree with you and have incorporated this suggestion in Abstract and our manuscript have been revised (highlighted sentences by yellow) as follows.

              (In line 19) “A total of 4 patients who needed peri-implant soft tissues reconstruction around dental implants received transplantation of atelocollagen-matrix (TERUDERMIS®) with MGCTs micronized by the tissue disruptor technique.

- Authors must include in the whole manuscript that this is a pilot case series.

We have revised whole manuscript including Title (highlighted sentences by yellow) as follows.

              (In line 3 and 4) “First-in-human study to investigate the safety assessment of peri-implant soft tissue regeneration with micronized-gingival connective tissue: A Pilot Case Series Study.”

              (In line 14) “This clinical study was a pilot case series, and the objective was to assess the safety and the preliminary efficacy of MGCTs on gingival tissue regeneration in dental implant therapy.”

              (In line 102) “… in dental implant treatment were evaluated in this pilot case series.

              (In this line 120)” The procedure of this pilot case series is summarized in Figure 1.”

              (In this line 355) “In conclusion, although there were only four patients in this pilot case series, …”

Introduction:

- Even though the authors affirmed they performed a language revision in the whole manuscript, very few modifications were present and many English use mistakes can be seen. Please, proof-read the manuscript again and edit the text accordingly.

We apologize for the inconvenience that we have caused you. As follows, our manuscript has been politely corrected by our colleagues whose native language is English.

              (In line 29) “Successful treatments in dental implants…”

              (In line 32) “…a concurrent decline in gingiva.”

              (In line 34)”… successful treatment in dental implants

              (In line 55) “potent regenerative and therapeutic potentials in the treatment of gingival defects.”

              (In line 74)”…, micro-grafts from adipose or dermal connective tissues have been shown to improve wound healing,…”

              (In line 88)”Therefore, survived mesenchymal cells derived from MGCTs promoted vascularization and re-epithelialization in a paracrine manner.

              (In line 107)”… (approval no. CRB7180001) and was registered with the Ja-pan Registry of Clinical Trials (trial no. jRCTs072180075)…”

              (In line 254) “The collagen sheet with micronized-gingiva is seated around the healing abutments.”

              (In line 268) ”… micronized-gingiva is seated around …”

              (In line 329)”Furthermore, GMMSCs possessed not only multipotent differentiation capacity, but also immunosuppressive and anti-inflammatory functions”

Materials and Methods:

- Legends from Figure 1 are missing.

We have added Legends from Figure 1 as follows.

              (In line 121) Figure1. The procedure of pilot study. All processes of micro-graft with MGCTs preparation and transplantation surgery were performed in a clean operation at the Nagasaki University hospital. The safety verification was evaluated in terms of the clinical course until 4 weeks after surgery.

- As requested in the last round of revision, please, use bullet points instead of numbers for the inclusion and exclusion criteria. The use of numbers can be confusing.

We thank you again for your appropriate comment. As follows, our manuscript has been corrected in Material and Methods section.

              (In line 124)

Patient Inclusion Criteria

・Dental implant treatment.

・Gingival defect or insufficient thickness (<2 mm).

・Oral health care and good plaque control.

・Aged 20 years or older, but younger than 75 years.

・Written, informed consent (IC) obtained from the patient him/herself.

・Intention and ability to visit a hospital.

・Intention to use barrier contraception during the study period.

Patient Exclusion Criteria

・Any malignancy or sepsis.

・Severe autoimmune or endocrinological disease; coagulation abnormality (PT<50% or outside the APTT range 23.5–42.5 seconds).

・Syphilis test/HBV antigen/HCV antigen/anti-HTLV-1 antibody/anti-HIV antibody positivity.

・Liver dysfunction (two biomarkers of liver function with concentrations outside the following range: AST 10–40 IU/L, ALT 5–45 IU/L).

・Pregnancy.

・Risk of allergy to the drug used in this study.

・Transmissible spongiform encephalopathy.

・Dementia.

・Metabolic bone disease or treated with bisphosphonates.

・Smoking habit.

・Severe periodontitis.

・Judged by the clinical investigator as inappropriate for this study.

- I still can see in Figure 1 the term “protocol treatment” instead of “treatment protocol.” Care must be taken during the revision process. The authors should be conscious and aware to avoid repetition of mistakes. Please, revise the WHOLE manuscript to the next round of revision.

We are deeply apologized for not accomplish it. We have revised Figure 1 and checked carefully whole manuscript.

- Again, very few data can be extracted from the proposed experiments. Considering this, I would include in the Discussion section the limitations of this study. The authors might say that this pilot case series is restricted to the safety assessment of the technique in solely 4 patients, and that more studies are necessary for the understanding of this possible graft.

We appreciate your suggestions, which help us to greatly improve the manuscript. We agree with you and have incorporated this suggestion in Discussion section.

              (In line 347) “Meanwhile, this pilot case series is restricted to the safety assessment of the technique in solely four patients and more studies are necessary for the understanding of this micro-graft with MGCTs. Therefore, the preliminary outcome of this study enables us to proceed to the next randomized controlled trial to assess the efficacy of the micro-graft with MGCTs including more patients, clinical assessment regarding health of peri-implant soft tissue, and longer follow up.

Results:

- Line 231: patients were submitted, not underwent. English revision is still necessary.

We appreciate the reviewer for the appropriate comment. Following your suggestion, we corrected the manuscript (highlighted sentences by yellow in line X) as follows. We hope these revised descriptions respond to your suggestions.

              (In line 231)

Case 1. A 67-year-old woman who had inadequate gingiva in both mandibular molar regions (Figure 4) was submitted to a one-stage implant, with the implant placed on 36, 37, 46, and 47 and a micro-graft. A piece of gingival connective tissue was harvested from the palatal mucosa of both maxillary tuberosities. The collagen sheet (size 10 × 20 mm2) with MGCTs covered the periosteum surrounding the implants. Four weeks after the surgery, there was an expanded region of gingiva at the buccal side, and this area was maintained 2 months after transplantation. Throughout the entire period, no AEs were observed.

              (In line 245)

Case 2. A 58-year-old woman who had less gingiva in the left maxillary premolar region (Figure 5) with the implants placed on 22, 24, and 25 was submitted to a second-stage surgery at 5 months post-first surgery. The gingival connective tissue was harvested from the palatal mucosa of the left maxillary tuberosity, and a collagen sheet (size 5 × 20 mm2) was seated around the healing abutments. The reconstructed gingiva was found surrounding the healing abutment at 4 weeks after surgery, and this region remained at 16 weeks. During the study protocol, no AEs were confirmed.

              (In line 258)

Case 3. A 71-year-old woman who had the implants placed on 25 and 26 was submitted to second-stage surgery at 6 months post-first surgery (Figure 6). There was little gingiva in the left maxillary premolar region. The non-keratinized gingiva was shifted to both buccal and palatal sides with split-thickness. From the palatal mucosa of the left maxillary tuberosity, the gingival connective tissue was harvested, and a collagen sheet (size 20 × 20 mm2) was seated around the healing abutments. Four weeks after surgery, the gingiva was obviously increased surrounding the healing abutment. Throughout the whole period, no AEs were observed.

              (In line 272)

Case 4. A 68-year-old man who had the implants placed on 32 and 42 was submitted to second-stage surgery at 7 months post-first surgery (Figure 7). The gingiva was observed at the alveolar crest line. The gingival connective tissue was harvested from the palatal mucosa of the right maxillary tuberosity, and a collagen sheet (size 10 × 20 mm2) was seated at the labial side of the healing abutments (32 and 42). The reconstructed gingiva was found on the labial side at 2 weeks after surgery. Four weeks after surgery, the gingiva was observed around the healing abutments. Throughout the entire period, no AEs were observed.

- From a personal standpoint, such a small sample size (4 patients), with such a limited analysis (only adverse events occurrence, which is very rare in Dentistry), with such a short evaluation time (4 weeks) cannot compose a full article publication. My vision is that the authors need to include more patients, include more parameters (maybe clinical assessment of peri-implant health, PD, BOP, KMW, etc.), and longer follow up (in four weeks not even the graft is fully recovered).

We are most appreciative of your constructive criticisms. We would like to proceed to the next RCT incorporating your meaningful suggestions. As above, we have revised in Discussion section.

              (In line 348) “Meanwhile, this pilot case series is restricted to the safety assessment of the technique in solely four patients and more studies are necessary for the understanding of this micro-graft with MGCTs. Therefore, the preliminary outcome of this study enables us to proceed to the next randomized controlled trial to assess the efficacy of the micro-graft with MGCTs including more patients, clinical assessment regarding health of peri-implant soft tissue, and longer follow up.

Discussion:

- I understand the authors and agree that your case series is the first step for the following RCT with the proposed method. However, we cannot trust in data provided from four cases and limited to a single safety assessment analysis. Moreover, the authors should clearly mention the limitations of the study in the Discussion section, as well as the recommendations for future investigations.

First of all, we would really appreciate your understanding the aim of this study and our perspectives. Following your advice, we have clearly mentioned the limitations of this study and added explanations regarding our perspectives. We hope these additional descriptions respond to your suggestions.

              (In line 348) “Meanwhile, this pilot case series is restricted to the safety assessment of the technique in solely four patients and more studies are necessary for the understanding of this micro-graft with MGCTs. Therefore, the preliminary outcome of this study enables us to proceed to the next randomized controlled trial to assess the efficacy of the micro-graft with MGCTs including more patients, clinical assessment regarding health of peri-implant soft tissue, and longer follow up.

Round 3

Reviewer 3 Report

Comments to authors

Dear authors,

Thank you for submitting the revised version of the manuscript entitled "First-in-human study to investigate the safety assessment of peri-implant soft tissue regeneration with micronized-gingival connective tissue: A pilot case series study" for potential publication in Medicines journal. The authors addressed all my comments thoroughly thereby improving their manuscript substantially. The manuscript can now be accepted for publication. During the final editing please correct the following minor points:

Line 29, please change "gingiva" to "keratinized tissue band". Please, be consistent along the manuscript replacing the “gingival” tissue around the implants by “peri-implant mucosa” (for example, gingiva on line 33, so on).

Legend Figure 3H, please add more information describing that the region was protected with a periodontal dressing, etc.

Table 1, please include a description.

Be consistent along the text with superscript numbers (mm3, mm square, mm3). The authors different notations, but you should use a pattern in the whole manuscript.

Thank you and congratulations.

Author Response

Dear authors,

Thank you for submitting the revised version of the manuscript entitled "First-in-human study to investigate the safety assessment of peri-implant soft tissue regeneration with micronized-gingival connective tissue: A pilot case series study" for potential publication in Medicines journal. The authors addressed all my comments thoroughly thereby improving their manuscript substantially. The manuscript can now be accepted for publication. During the final editing please correct the following minor points:

We would like to express our appreciation to the reviewer for her/his insightful comments, which have helped us to substantially improve the manuscript. We have reflected your comments and revised the manuscripts. We hope these additional revises respond to your suggestions.

Line 29, please change "gingiva" to "keratinized tissue band". Please, be consistent along the manuscript replacing the “gingival” tissue around the implants by “peri-implant mucosa” (for example, gingiva on line 33, so on).

We agree with you and have incorporated this suggestion in our manuscript. We have revised (highlighted sentences by yellow) as follows.

              (In line 14)”…keratinized tissue band.

              (In line 16)”… on peri-implant mucosa regeneration.”

              (In line 24)”… promising for peri-implant mucosa reconstruction…”

              (In line 29) “…an adequate keratinized tissue band to…”

              (In line 30)”…, since an insufficient keratinized tissue band results in…”

              (In line 33)”decline in keratinized tissue band.

            (In line 34) “…reconstruction or augmentation of adequate peri-implant mucosa is indispensable for…”

              (In line 41)” inadequate width and thickness of peri-implant mucosa

              (In line 47)” width and thickness of peri-implant mucosa

              (In line 56)” in the treatment of peri-implant mucosa defects”

              (In line 65)”of peri-implant mucosa

              (In line 126)” Keratinized tissue band defect”

              (In line 169)”the peri-implant mucosa surrounding the implants was shifted apically…”

              (In line 177)” the keratinized tissue band area”

              (In line 180) “The peri-implant mucosa surrounding the implants

              (In line 232)”… who had inadequate keratinized tissue band…”

              (In line 237)”…an expanded region of peri-implant mucosa at the buccal side…”

              (In line 241)” The keratinized tissue band was located…”

              (In line 243)”… The region of peri-implant mucosa is expanded…”

              (In line 244)” The peri-implant mucosa region…”

              (In line 246)” who had less peri-implant mucosa in the left maxillary…”

    (In line 250)” The reconstructed peri-implant mucosa was found…”

              (In line 254)” The keratinized tissue band is located at the alveolar crest…”

              (In line 255)” The peri-implant mucosa is observed…”

              (In line 256)” The area of peri-implant mucosa is maintained…”

              (In line 261)”There was little keratinized tissue band in the left maxillary premolar region.”

              (In line 261)” non-keratinized tissue band is shifted…”

              (In line 265)” the peri-implant mucosa was obviously increased…

              (In line 268)” The keratinized tissue band is almost disappeared.”

              (In line 268)” Non-keratinized tissue band is shifted…”

              (In line 270) “The peri-implant mucosa is found…”

              (In line 274)” The keratinized tissue band was observed…”

              (In line 278)” The reconstructed peri-implant mucosa was found…”

              (In line 279) “…, the peri-implant mucosa was observed…”

              (In line 282)” The keratinized tissue band is…”

              (In line 283)” The keratinized tissue band is shifted…”

              (In line 283)” The peri-implant mucosa is observed…”

              (In line 289)” thickness of the peri-implant mucosa…”

              (In line 294)” for peri-implant mucosa regeneration…”

              (In line 300)” for peri-implant mucosa reconstruction”

              (In line 318)” in peri-implant mucosa reconstruction…”

              (In line 321)” on peri-implant mucosa regeneration…”

              (In line 323)” …micro-graft supported peri-implant mucosa augmentation…”

              (In line 326)” contribute to peri-implant mucosa construction or augmentation…”

              (In line 338)” for peri-implant mucosa regeneration…”

              (In line 339)” there was little peri-implant mucosa…”

              (In line 341)” reconstructed peri-implant mucosa was obviously found…”

              (In line 343)” the regeneration of peri-implant mucosa.”

              (In line 345)” for peri-implant mucosa augmentation…”

              (In line 359)” for peri-implant mucosa regeneration…”

Legend Figure 3H, please add more information describing that the region was protected with a periodontal dressing, etc.

Following your suggestion, we have revised our manuscript. For additional information, our manuscript is revised in Legend Figure 3H (highlighted sentences by yellow) as follows.

              (In line 182) H; The region was protected with a periodontal dressing after surgery.

Table 1, please include a description.

I apologize for not explaining it clearly. We have included a description in Table 1 (highlighted sentences by yellow) as follows.

              (In line 231) List of the patients who participated in the clinical study.

.

Be consistent along the text with superscript numbers (mm3, mm square, mm3). The authors different notations, but you should use a pattern in the whole manuscript.

We appreciate your appropriate comment. According to your kind suggestion, we use a pattern in the whole manuscript.

              (In line 92) “approximately 3 mm3

              (In line 150) “approximately 3 mm3

              (In line 250)” size 5 × 20 mm2

              (In line 264) “size 20 × 20 mm2

              (In line 285)” approximately 3 mm3